# Axially-Anisotropic Hierarchical Grating 2D Guided-Mode Resonance Strain-Sensor

**DOI:** 10.3390/s19235223

**Published:** 2019-11-28

**Authors:** Sachin Babu, Jeong-Bong Lee

**Affiliations:** Department of Electrical & Computer Engineering, University of Texas at Dallas, Richardson, TX 75080, USA

**Keywords:** guided-mode resonance, 2D grating, refractive index, diffraction grating, quality factor

## Abstract

Guided-mode resonance strain sensors are planar binary gratings that have fixed resonance positions and quality factors decided by material properties and grating parameters. If one is restricted by material choices, the quality factor can only be improved by adjusting the grating parameters. We report a new method to improve quality factor by applying a slotting design rule to a grating design. We investigate this design rule by first providing a theoretical analysis on how it works and then applying it to a previously studied 2D solid-disc guided-mode resonance grating strain sensor design to create a new slotted-disc guided-mode resonance grating design. We then use finite element analysis to obtain reflection spectrum results that show the new design produces resonances with at least a 6-fold increase in quality factor over the original design and more axially-symmetric sensitivities. Lastly, we discuss the applicability of the slotting design rule to binary gratings in general as a means of improving grating performance while retaining both material and resonance position choices.

## 1. Introduction

Strain sensing has numerous applications, from concrete structures [1] to biomechanics and robotics [2], in which the displacement of a deformable material under a force needs to measured. While there are several physical phenomena (piezoresistivity, piezoelectricity, capacitance) that can be used for measuring strain, of which piezoresistivity is currently the dominant form [3], a large number of optics-based strains sensors are being developed due to its smaller size, low power consumption, high sensitivity, large bandwidth, biocompatibility, and immunity to electromagnetic interference [4]. The typical optics-based strain sensor is the fiber optic strain sensor, of which there are many variants, each of which exploit different optical phenomenon such as attenuation, fluorescence, luminescence, interference, to name a few [4]. Of these, fiber optics using interference, specifically Bragg gratings, are the most prevalent. While fiber Bragg gratings are highly sensitive—capable of microstrain resolution—they have high stiffness that limit range of operation and require direct fiber optic coupling to a detector.

Guided-mode resonance (GMR) is a phenomenon that occurs when electromagnetic radiation incident upon a binary dielectric grating becomes coupled to the leaky (radiative) waveguide modes of that grating [5,6]. Depending on the grating design, these leaky modes allow resonance bandwidths to be quite sharp, often going from near-unity transmission to near-unity reflection (or vice versa) over a narrow range of wavelengths. The design criteria to achieve such resonances are based on grating parameters and material permittivity choices. Given the wide range of fabrication feature sizes and a large selection of dielectric materials to choose from, GMR devices can be fabricated to operate over a variety of wavelength spectra that are of interest.

GMR theory [7,8,9] was initially applied to design all-dielectric high quality factor optical filters out of a single thin film binary grating with a substrate and superstrate. The theory was eventually implemented to fabricate filters that were designed to have reflection (or transmission) resonance peaks in the visible [10], infrared [11,12], and gigahertz ranges [13]. The applicability of GMR to such a wide range of wavelengths is due to the geometric scalability of the grating design parameters [14]. More GMR applications started appearing in areas involving vertical-cavity surface-emitting lasers (VCSELs) [15,16], tunable filters [17], and microscale optical elements such as absorbers [18] and focusing lenses [19]. GMR has also found application in the area of solar technology, in which gratings are designed to trap incident solar rays [20,21].

In the area of biosensing, GMR has been used with great success to detect a variety of analytes. Based on the principle that the analyte deposited on the GMR grating will slightly alter the refractive index consequently resulting in a measurable shift in the resonance peak, GMR biosensing can be used to detect extremely small quantities of analyte. It has been used to detect a variety of analytes from pollutants in water [22] to proteins such as biotin [23], biomarkers for ovarian cancer [24] or biomarkers for acute myocardial infraction [25]; it also has been used to detect various solvents such as methanol, ethanol, 2-propanol, and cyclohexane [26].

More recently, the GMR phenomenon was found to be applicable towards creating high-sensitivity pressure and strain sensors [27,28]. These sensors show great promise for measuring strains of soft materials, such as skin or muscle, offering a larger range of operation and more remote methods of measurement. Unlike most GMR-based devices, which typically use rigid crystalline dielectric substrates, GMR pressure and strain sensors require more compliant substrates to allow pressure or strain sensing over large ranges. Polydimethylsiloxane (PDMS), a polymer with high transmissivity [29,30] and high elasticity [31], has been used as a compliant substrate for implementing various GMR pressure and strain sensors [32,33,34]. PDMS was initially suggested as a substrate for microfluidics and biotechnology applications [35] and has since been ubiquitously used in those fields. Thus, there is great advantage in developing GMR-based sensors using this material.

Foland and Lee [36,37] were the first to design and fabricate a 2D highly compliant GMR strain sensor by embedding a square array of TiO_2_ discs in PDMS elastomer. The planar array of TiO_2_ discs and PDMS form a binary grating with PDMS as the substrate and air as the superstrate to create a 2D GMR grating. When light from a broadband source is normally incident on the grating, two resonant peaks are reflected back. In order to generate two non-overlapping near-unity reflection peaks, the pitches between the discs along the orthogonal planar axes (*Z* and *Y*, in this case) are required to be asymmetric (*Λ_Z_* ≠ *Λ_Y_*). If a uniform strain is applied along one axis, say the *Z*-axis, there is a uniform displacement in the grating pitch whose grating vector is along the *Z*-axis, and this displacement results in a shift in the resonant peak for that axis, called the *Z*-axis peak. A strain applied along the *Y*-axis similarly results in a shift in the *Y*-axis peak. Considering thin film TiO_2_ is a crystalline dielectric with elastic modulus of about 65 GPa [38], it has a negligible strain when compared with PDMS, which has an elastic modulus of about 750 kPa [39]. Thus, under applied strain, of the grating parameters (*Λ_Z_*, pitch along *Z*-axis; *Λ_Y_*, pitch along *Y*-axis; *d*, disc diameter; *t*, disc thickness; disc permittivity *n_TiO2_* = *n_H_* = 2.35; and substrate permittivity *n_L_* = *n_PDMS_* = 1.4) one should expect only the GMR grating pitches to change. Under *Z*-axis strain, the *Z*-axis pitch, *Λ_Z_*, should increase; while under *Y*-axis strain, the *Y*-axis pitch, *Λ_Y_*, should increase. Since PDMS is highly compliant, with a Poisson ratio nearly 0.5 [39], the strain applied to one axis does not affect the other much; though there is some discernable shift. Consequently, the two reflection peaks can be said to move largely independently of each other, allowing simultaneous measurements of axial strains.

The device, however, suffers from a relatively low quality factor (*Q* < 100), and unequal axial sensitivity due to the asymmetric pitch requirement. To improve the quality factor of this sensor, one can either pick different materials or adjust the grating design [8]. For this grating, one cannot choose to pick different materials, since the elastic and optical transparency of PDMS is required for the large strain sensing operating range and GMR, respectively; and an alternative dielectric material with a high and stable index of refraction over the desired wavelength range similar to TiO_2_ is hard to find. Thus, one must approach improving the quality factor of this GMR grating strain sensor by adjusting the grating design.

There are not too many parameters to adjust in a 2D square-array-of-discs grating design. Apart from the radius of the discs, the thickness of the discs, replacing discs with a different shape, changing grating pitch, or eliminating layers [40], there is no other parameter one can fine tune. There are several papers that report improving GMR quality factor through adjusting these grating parameters. Fan et al. theoretically investigated reducing the radii of holes of a slab photonic crystal and found the quality factor generally increases by at least a factor of 23 if the radius is reduced to a quarter of the original [41]. However, Pottier et al. reported that while reducing radius does increase quality factor, it also severely reduces the peak intensity [42]. Fattal et al. simulated the role of slab grating thickness on quality factor and found that reducing the thickness of the grating (from 200 to 10 nm) improved the line width significantly, but their result aren’t easy to quantify; furthermore, they admit such grating thicknesses would be too difficult to fabricate over large areas with current limitations of technology [43]. Andreani et al. show that replacing a circular shape with a triangular shape increases the *Q*-factor by less than 2% [44], essentially a negligible improvement in quality factor. Lastly, increasing the grating pitch is a reasonable method to improve quality factor, however adjusting grating pitch also affects the resonance positions dramatically and can push the grating into resonance–free ranges [8]. If one requires to place resonances in easily-measurable wavelength ranges such as in the visible or NIR regimes, adjusting the grating pitch to achieve higher quality factor is not an option.

One approach to altering grating parameters of GMR gratings not found in literature is to replace the high-index disc itself with a grating. This grating-within-a-grating, or hierarchical grating, approach offers a method to improve the quality factor of a GMR grating through lowering the effective refractive index and provides a new grating design parameter that allows one to control the duty cycle of the sub-grating. By adjusting this duty cycle one can control the proportion (or fill factor) of the sub-grating allowing one to fine-tune the refractive index of the high-index part of the grating while keeping all other aspects of the grating the same. This hierarchical grating approach can be codified as a design rule (called the “slotting design rule”) wherein the high index part of a GMR grating is slotted to form a grating. The hypothesis is that the slotting design rule, when applied to a GMR grating, should improve the quality factor; however there maybe unexpected benefits and side effects. In this paper, we apply this slotting design rule to a 2D guided mode strain sensor design by Foland and Lee [36] to create an axially-anisotropic hierarchical 2D GMR grating strain-sensor design and show that it has an improved quality factor (by a factor of 6) and more symmetric axial sensitivity.

## 2. Methods

### 2.1. Design of Slotted-Disc GMR Filters

The impetus to use slotted disc design came from trying to understand the role of grating parameters and material permittivities on the quality factor of the resonance peaks. The quality factor, *Q*, is defined as:(1)$$Q\equiv \;\frac{f_{r}}{\Delta f}=\frac{\lambda_{H}\lambda_{L}}{\lambda_{r}}\frac{1}{\Delta \lambda }\propto \frac{1}{\Delta \varepsilon }$$
where *ƒ_r_* is the resonance frequency; *Δƒ* is the resonance width (the full-width half-maximum); *λ_r_* is the resonance wavelength or position; and *Δλ = λ_H_ − λ_L_*, where *λ_H_* and *λ_L_* are the wavelengths on either side of the resonance wavelength that are at half the peak reflectance values of the resonance peak.

According to works done by Magnusson et al. and others [6,7,8,9] the linewidth of a resonance peak, *Δλ*, is proportional to the modulation index, $\Delta \varepsilon =\;\varepsilon_{H}-\varepsilon_{L}=\sqrt{n_{H}}-\sqrt{n_{L}}$, and this relationship is noted on the right hand-side of Equation (1). Thus, one can improve *Q*, for a given resonance position, *λ_r_*, by lowering the modulation index. One could attempt to lower the modulation index by selecting two different grating materials that are closer in permittivity values or by creating nano-composites [45]. However, this introduces additional steps to the process flow and may require additional tools and testing. With the slotting design rule one would retain the same material choices and only have to alter a binary grating design.

Additionally, changing materials may not always be an option. Some GMR strain sensors are limited to selected materials due to restrictions of resonance wavelength range, and/or requirements of transparency, elasticity, and biocompatibility, such as the 2D GMR strain sensor. For these reasons, there is value in finding a solution that lowers the modulation index while retaining material choices.

The solution can be found by studying the effective index of refraction of dielectric gratings, which, according to effective medium theory [46], is the average permittivity that can be assigned to the grating for modes with wavelengths much greater than the grating pitch. One can now consider creating a slotted disc by replacing the high index part of the larger grating with a sub-grating composed of alternating slivers of the same high permittivity and low permittivity material. Applying this concept, one can replace the solid TiO_2_ discs (Figure 1a) with that of a TiO_2_/PDMS disc grating or slotted TiO_2_ discs (Figure 1b). The modulation index can then be tuned over the range of *n_L_* to *n_H_* by adjusting the duty cycle *q* of the disc grating. Lowering the duty cycle reduces the modulation index, which should result in an improvement of the quality factor. Thus, one can now improve quality factor by having some measure of control on the effective index of refraction of the high-index part of binary gratings.

### 2.2. AxiallyAnisotropic Refractive Index

Slotted-disc gratings do not have an isotropic index of refraction, like the solid-disc gratings do. They instead have an axially-anisotropic index of refraction, and it can be understood by studying effective medium theory (EMT). In EMT, the slotted disc can be treated as a binary subwavelength grating, which has been well studied [47,48,49]. For a binary subwavelength grating with pitch *S*, fill factor *q*, and thickness *t*, with index of refractions *n_H_* and *n_L_* (Figure 2), EMT gives the approximate values of the effective indices of refractions to be [48]:(2)$$n_{\left|\right|}={\left[n_{H}^{2}q+n_{L}^{2}\left(1-q\right)\right]}^{1/2}$$
(3)$$n_{⊥}={\left[(1/n_{H}^{2})q+(1/n_{L}^{2})\left(1-q\right)\right]}^{-1/2}$$
where ‖ and ⊥ denotes the electric field, E, of the incident light parallel to the binary grating and perpendicular to the binary grating, respectively. In this work, the E-field is always kept perpendicular to the grating vector. 

While there are more accurate methods of calculating effective permittivities, Equations (2) and (3) provide estimates accurate enough for designing subwavelength gratings. Lalanne et al. [46] do provide a more accurate method for calculating the effective permittivities using 2D-RCWA (rigorous coupled wave analysis), which was confirmed by Kikuta et al. [50], however it would be quite difficult to describe geometrically-complex 3D structures such as the slotted-disc gratings and implement it in 2D RCWA code.

Lalanne et al. [51] also found that as the thickness-to-wavelength ratio, *t/λ*, approaches zero the values of the actual refractive index deviates rapidly from those predicted by Equations (2) and (3). However, their analyses show that for 2D gratings with *t/λ* values roughly 0.125 and above, the effective permittivities are very close to that predicted by Equations (2) and (3). For our device, the operating wavelength range is 600 nm–1100 nm, and *t* is 200 nm, giving us a minimum *t/λ* ratio of about 0.33, making Equations (2) and (3) nearly as accurate as the 2D-RCWA method. For these reasons, there is no need to use the 2D-RCWA method and one can chose to use Equations (2) and (3) to calculate effective index of refraction for subwavelength gratings. Lastly, while Equations (2) and (3) are meant for rectangular-shaped binary gratings, it can be used as a rough measure of the expected effective permittivities of the disc grating for the purposes of choosing an optimal *q*.

### 2.3. Choosing q and S

Figure 2b shows how one can choose a *Δn* based on a choice of fill-factor *q*. *q* = 0.5 was chosen in order to have a large difference between *n_‖_* and *n*_┴_ (these are the axially anisotropic indices of a slotted disc.). Having such a large difference is needed to create distinct non-overlapping resonance peaks. This is because the closer *n_‖_* and *n*_┴_ are to each other, the closer the resonance positions are to each other, making them harder to resolve. Note that the anisotropic refractive indices (*n_‖_* = 1.93 and *n*_┴_ = 1.70) are lower than *n_TiO2_* = 2.35, resulting in two modulation indices, *Δε_‖_* and *Δε*_┴_, that are lower than that of the original solid-disc grating, and, as a result of the lowering, should produce resonance peaks that have higher quality factor than the original grating. Note that the *Δ*ε*_‖_* applies to the axial direction parallel to the larger grating (*Y*-axis) and *Δε*_┴_ applies to the axial direction perpendicular to the larger grating (*Z*-axis) (see Figure 1b). For a solid-disc GMR filter device having a disc diameter of 420 nm, a slotting pitch *S* = 120 nm was chosen, and with *q* = 0.5, *qS* = 60 nm. This more than meets the criteria of having the slotted-disc grating pitch be less than half the wavelength of the incident radiation (600 nm–1100 nm) such that effective medium theory applies [48].

## 3. Results

### 3.1. 3D FEA Simulation

From the slotting parameters chosen above, one can now model the solid-disc and slotted-disc 2D GMR gratings using finite element analysis (FEA) to obtain reflection spectra. This method was previously shown to be equivalent to the well-established RCWA method [52,53] for GMR gratings in Foland et al. [54]. All models of both 2D GMR gratings for this paper were implemented using COMSOL^®^ (COMSOL Inc., Burlington, MA, USA). In the models, a unit cell of the 2D GMR grating was implemented using Floquet periodicity for axial boundary condition matching. The unit cell was set up as a two-port system, with an input port acting as a source of TE (transverse electric) or TM (transverse magnetic) monochromatic light incident along the normal of the 2D grating plane from the top, and an output port of transmitted waves exiting from the bottom (Figure 3). After assigning material properties to the grating structures of the unit cell, and simulating this two-port system over a range of wavelengths, the |*S_11_*|-parameter was retrieved over the range of wavelengths (i.e., the reflectance spectrum of the device).

Several FEA models were constructed based on the parameters noted in Figure 1 caption for both the solid disc and slotted disc devices. Some of these models are rest models, which represent 0% strain, and other are strained models, which represent a unit cell with increased pitch due to the applied strain in either in the *Z*- or *Y*-axis direction, but not both simultaneously. In total, ten strained models were created: five models to represent 5%, 10%, 15%, 20% and 25% strains along the *Z*-axis, and another five to represent strains along the *Y*-axis. Of the two materials, TiO_2_ experiences negligible strain when compared with PDMS. For solid-disc grating under strain, the PDMS surrounding the discs will experience the strain; thus, the only grating parameters affected by strain are the GMR grating pitches (*Λ_Z_* and *Λ_Y_*). This effect was confirmed by simulation and experiment by Foland et al. [36] (our previous work). For slotted-disc grating under strain, both the PDMS surrounding and in-between the slotted-discs will experience the strain; thus, the grating parameters affected by strain are the GMR grating pitches (*Λ_Z_* and *Λ_Y_*) and the sub-grating pitch (the duty cycle, *q*). The strain experienced by PDMS within the slotted-discs is simulated using COMSOL^®^’s stress-strain module (Figure 4a). The simulations show that the effect of the strain increases the spacing between the slots, an example of which is shown in Figure 4b. Using these stress-strain simulations, the new spacings of the slotted-discs for a given strain were calculated and implemented in to the 2-port FEA model for that specific strain (Figure 4c,d) as the new sub-grating pitch.

The reflectance spectra from both the rest and strained models are plotted in Figure 5 (a,b for solid-disc; c,d for slotted-disc) for only the 0%, 5%, 15% and 25% strains along both *Z* and *Y* axes; the 10% and 20% plots were excluded for clarity. Note that ‘*Y*-peak’ denotes the peak that shifts under *Y*-axis strains, and ‘*Z*-peak’ denotes the peak that shifts under *Z*-axis strains.

### 3.2. Sensitivity Study Results

The sensitivity vs. strain plots were derived from COMSOL^®^ simulation reflectance plots for 0–25% strain in steps of 5% strain. The sensitivity was calculated by taking the ratio of the change in resonance peak position to the change in strain. Figure 6a presents the results for the solid-disc GMR grating design; Figure 6b presents the results for the slotted-disc GMR grating design.

For solid-disc grating, under *Z*-axis strain, the *Z* peak has an average sensitivity of 6.34 nm/% with a standard deviation of 0.31 nm/%, while the *Y* peak has an average of −0.23 nm/% with standard deviation of 0.12 nm/%. Under *Y*-axis strain, the *Y* peak has an average sensitivity of 4.64 nm/% with a standard deviation of 0.25 nm/%, while the *Z* peak has an average of −0.64 nm/% with standard deviation of 0.22 nm/%.

For slotted-disc grating, under *Z*-axis strain, the *Z* peak has an average sensitivity of 5.19 nm/% with a standard deviation of 0.14 nm/%, while the *Y* peak has an average of −0.35 nm/% with standard deviation of 0.09 nm/%. Under *Y*-axis strain, the *Y* peak has an average sensitivity of 5.67 nm/% with a standard deviation of 0.13 nm/%, while the *Z* peak has an average of −0.59 nm/% with standard deviation of 0.12 nm/%.

### 3.3. Quality Factor Study Results

The quality factor vs. strain plots (Figure 6c,d) were derived from COMSOL^®^ simulation reflectance plots for 0%–25% strain in steps of 5% strain. The quality factor is found by using a peak finding algorithm to find the wavelengths of any peaks with above 50% reflectance. After finding the peak wavelengths, the algorithm uses the reflectance spectrum data again to determine the full width half maximum (FWHM), or *Δλ = λ_H_ − λ_L_*, at the peak locations. From the peak wavelength and the FWHM one can calculate the quality factor of the peak using Equation (1).

For the solid-disc grating, under *Z*-axis strain, Figure 6c shows the *Z*-peak quality factor starts close to 30 at 0% strain, and tends to decrease with strain. The *Y*-axis peak quality factor rises for 5% strain, dropping slightly at 10% strain. At 15% strain, there is a dip in the quality factor, which may look like a simulation artifact, but it is due to the overlap of the *Y*-axis peak and *Z*-axis peak coinciding resulting in peak widening. As it goes from 15% strain toward 20% strain, the peaks move out of the overlap and their respective quality factors rise again. Under *Y*-axis strain, the *Y*-axis peak quality factor is 40 at rest and monotonically increases in value from 0% strain to 25% strain, except for the dip at around 15% strain due to the overlap.

For the slotted-disc grating, under *Z*-axis strain, Figure 6d shows the *Z*-peak quality factor is about 185, around a factor of 6 increase above the quality factor for the same in peak for the solid-disc grating. As strain is increased, the quality factor drops at 10% strain. There is a sudden rise in the quality factor to 220 at 15% strain before it drops again to below 200 for 20% and 25% strains. The *Y*-peak, under *Z*-axis strain, starts, at 0% strain, with a quality factor at 240, about a factor of 6 increase above the quality factor for the same peak for solid-disc grating, and monotonically increases, reaching 302 at 25% strain. Under *Y*-axis strain, the *Y*-peak, at 0% strain, has a quality factor of about 240 also, that rises slightly for 5% strain, then falls slightly for 10% strain, before rising in value as strain increases from 10% to 25%. Under the same *Y*-axis strain, the *Z*-axis peak quality factor is 185, at 0% strain, lowers slightly at 5% strain, rises slightly at 10% strain, lowers slightly again at 15%, before rising for the remaining strains.

## 4. Discussion

The slotting design rule was hypothesized to improve the quality factor of the 2D GMR grating without having to pick new materials or change grating pitch (to preserve resonance position). The simulation results clearly show that, under no strain, there is at least a 6-fold improvement in the quality factor of all resonance peaks simply by applying the slotting design rule to the solid-disc GMR grating strain sensor design. As mentioned in the theory sections, the slotting design rule works by lowering the effective index of refraction of the high index part of the 2D GMR grating, reducing the modulation index, thereby increasing qualify factor. However, there are some additional features that manifest from applying this design rule to the 2D GMR gratings.

Recall that for the solid-disc grating one is required to have an asymmetric pitch (*Λ*_Z_ ≠ *Λ_Y_*) to generate two distinct resolvable peaks [36]. This is because the high index part of the solid-disc GMR grating has the same index of refraction in both planar-axial directions. If the pitches were kept the same, the resonance would occupy the same position since both axial directions would have the same modulation index. By applying the slotting design rule to the solid-disc 2D GMR grating, one finds this axial-pitch asymmetry requirement is no longer needed. The slotting rules creates an axially-anisotropic effective index of refraction (i.e., the index of refraction experienced by light traveling along the *Z*-axis direction is slightly different from that experienced travelling along the *Y*-axis direction) out of the high index part of the GMR grating. Thus one can now keep the grating pitches is symmetric (*Λ_Z_* = *Λ_Y_*) and rely on the axially-anisotropic index of the high-index part of the grating to create two resolvable resonance peaks.

When strain is applied, differing behaviors are observed for sensitivity and quality factor. Sensitivity remains largely constant, not varying much in value for strains from 0% to 25% for both solid-disc and slotted-disc gratings. Quality factor, however, is drastically affected by strain. Part of this behavior is due to the close positions of the two resonance peaks, which move in opposite directions under strain, as noted by the arrow in the plots of Figure 4 indicating the motion of the peaks under strain. As a consequence, over a small region of strains the peaks overlap, becoming broader, and at some point overlap completely to become unresolvable. This is the point where the quality factor dips significantly before rising once again as the peaks continue their motion under strain out of overlap. This overlap feature, however, is common to both the solid-disc and slotted-disc 2D GMR grating so cannot be a feature brought about by the slotting design rule. Once past the region of strain that contain the overlap, the quality factor tends to rise for the *Y*-axis peak under *Y*-axis strain, and generally drop for the *Z*-axis peak under *Z*-axis strain for both solid-disc gratings and slotted-disc gratings. Since this drop in quality factor for the *Z*-axis peak occurs for both gratings, it also cannot be a feature brought about by the slotting rule. Thus, while the slotting design rule does increase quality factor overall, it will be affected by other properties and behaviors of the 2D GMR grating strain sensor.

Lastly, one should note that the slotting design rule should be applicable to any grating design that involves binary gratings, which has a high index part and a low index part. The purpose of slotting is essentially to create a parameter by which to have some control over the index of refraction of the high index part of the GMR grating. That parameter is *q*, which allows control over the duty cycle of the sub-grating that replaces the high index part, and allows a range of values between the high index and low index values. As discussed in the theory part, when *q* is reduced, this has the effect of lowering the modulation index, which increases the quality factor. This effect is independent of material choice or grating design, and is applicable to all binary gratings. Thus, the slotting design rule can be considered as a universal design rule to improve the quality factor of any binary grating.

## 5. Conclusions

A slotting design rule was investigated as a method to improve the sensitivity and quality factor of a 2D solid-disc GMR grating strain sensor. Effective medium theory was utilized to demonstrate the expected improvement in quality factor through reducing modulation index. Grating theory was utilized to show how the slotting design rule can be used as a method for controlling the modulation index, thus providing a means of improving quality factor. The theory was used to design a 2D slotted-disc GMR grating strain sensor. FEA simulations were performed and reflectance spectra were analyzed to show that the 2D slotted-disc GMR strain sensor produces two non-overlapping resonance peaks without needing asymmetric pitches, has more axially symmetric sensitivities, and yields a 6-fold increase in the quality factor at rest for both resonance peaks. It was also shown, that under increasing strain, the sensitivities remain stable, while the quality factor generally tends to vary dramatically. Lastly, the slotting design rule was shown to be applicable to any binary grating and can be used to improve the performance of such grating in a wide variety of applications.

## Figures and Tables

**Figure 1 sensors-19-05223-f001:**
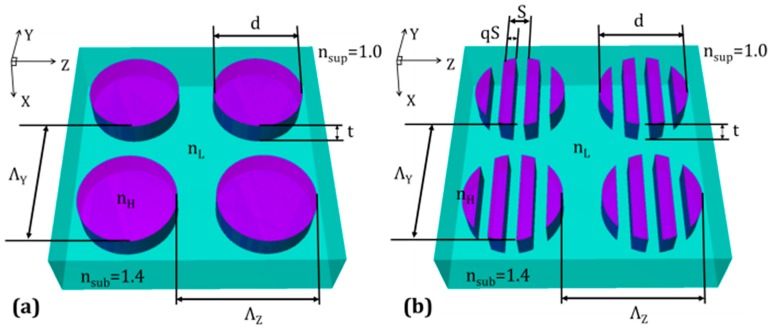
Design of 2D GMR grating strain sensor. (**a**) Solid-disc GMR reflection filter, consisting of discs of TiO_2_ embedded in PDMS; *Λ_Z_* = 560 nm, *Λ_Y_* = 480 nm, *t* = 200 nm, *d* = 420 nm, *n_H_* = 2.35, *n_L_* = 1.4, *n_sup_* = 1.0, *n_sub_* = 1.4. (**b**) Slotted disc GMR reflection filter, a grating-within-a-grating device, consisting of slotted-discs of TiO_2_ embedded in PDMS; *Λ_Z_* = *Λ_Y_* = 480 nm, *t* = 200 nm, *d* = 420 nm, *S* = 120 nm, *q* = 0.5, *n_H_* = 2.35, *n_L_* = 1.4, *n_sup_* = 1.0, *n_sub_* = 1.4.

**Figure 2 sensors-19-05223-f002:**
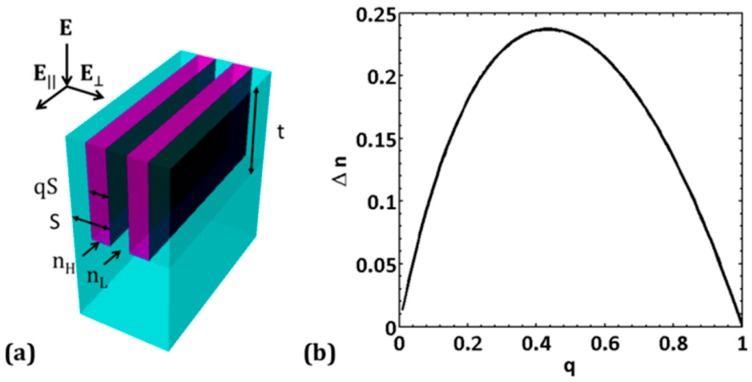
(**a**) Binary grating with PDMS substrate and air superstrate. The grating parameters are *S* = 120 nm, *q* = 0.5, *n_H_* = 2.35, *n_L_* = 1.4, *n_sup_* = 1.0, *n_sub_* = 1.4. (**b**) Plot of *Δn* = (*n_‖_* − *n*_┴_), vs. *q*; for *q* = 0.5, *n_‖_* = 1.93 and *n*_┴_ = 1.70, and *Δn* = 0.23.

**Figure 3 sensors-19-05223-f003:**
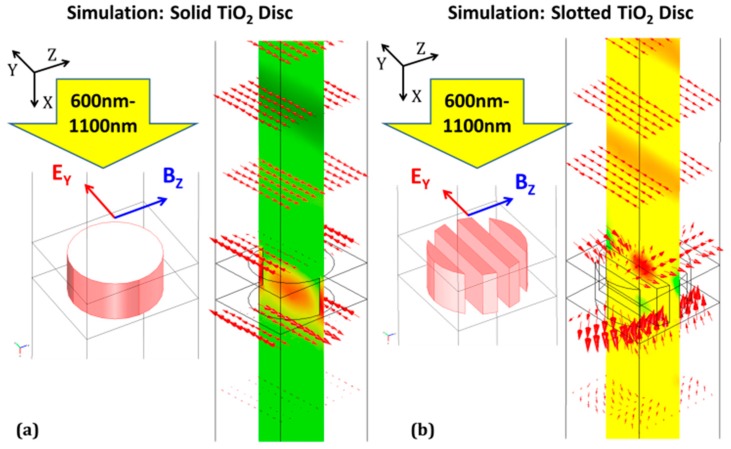
COMSOL^®^ simulation of (**a**) solid disc GMR filter device using parameters in Figure 1a (**b**) slotted-disc GMR filter device using parameters in Figure 1b.

**Figure 4 sensors-19-05223-f004:**
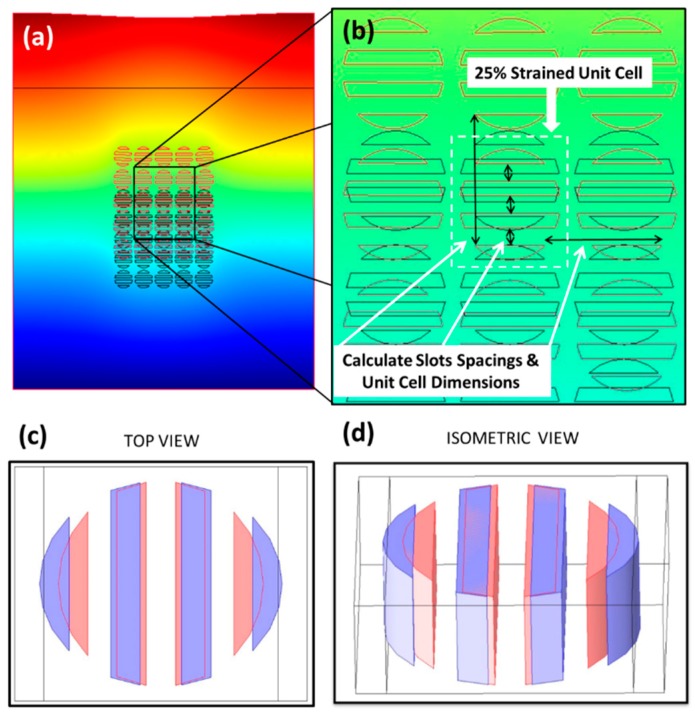
(**a**) COMSOL^®^ stress-strain simulation of slotted-disc for 25% *Z*-axis strain; (**b**) close-up view of center 25% *Z*-axis strained unit cell; resting slotted disc (pink) and 25% *Z*-axis strained slotted disc (purple) in (**c**) top view, and (**d**) isometric view, used for the 2-port FEA model.

**Figure 5 sensors-19-05223-f005:**
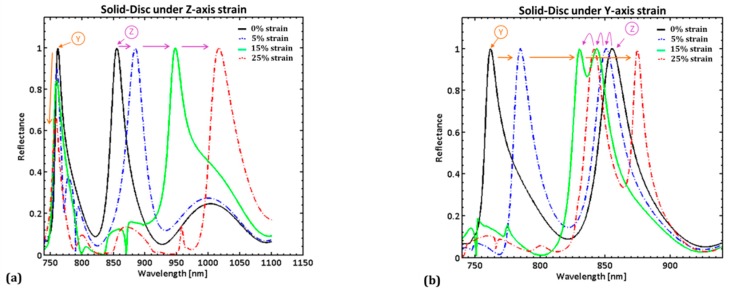
Reflection spectrum obtained from COMSOL^®^ simulations using the RF module. The ‘*Y*-peak’ denotes the peak that shifts under *Y*-axis strains, and ‘*Z*-peak’ denotes the peak that shifts under *Z*-axis strains. (**a**) solid-disc under *Z*-axis strain; Note the drop in *Y*-peak reflectance. (**b**) solid-disc under *Y*-axis strain; note the *Y*-peak partially overlaps the *Z*-peak around 15% strain, and moves past it at 25% strain. (**c**) slotted-disc under *Z*-axis strain; note the drop in reflectivity of the *Y*-peak. (**d**) slotted-disc under *Y*-axis strain; there is a significant drop in *Z*-peak reflectance.

**Figure 6 sensors-19-05223-f006:**
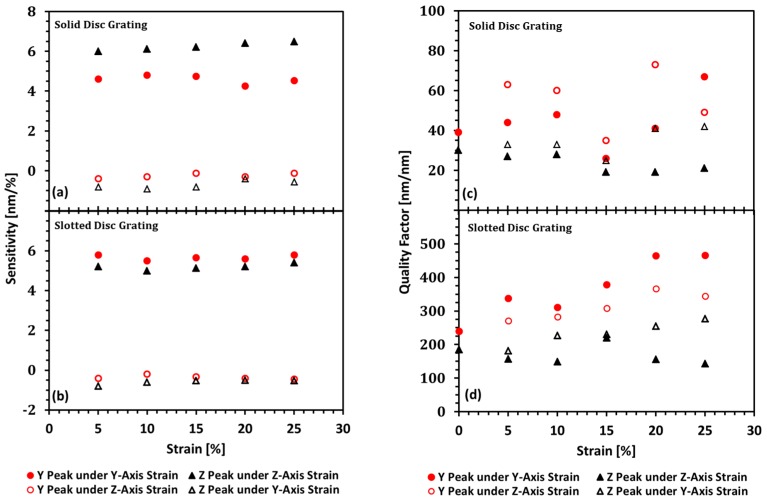
(**a**) Sensitivity vs. strain for solid-disc grating. (**b**) Sensitivity vs. strain for slotted-disc grating. (**c**) Quality factor vs. strain for solid-disc grating. (**d**) Quality factor vs. strain for slotted-disc grating.

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
