# Peer review of "Axially-Anisotropic Hierarchical Grating 2D Guided-Mode Resonance Strain-Sensor"

_sensors, 2019, doi:10.3390/s19235223_

Round 1

Reviewer 1 Report

The paper reports on theoretical simulation of guided mode resonance filters as strain sensors.  The structure is well know and have been applied to tunable filtering, refractive index sensing and temperature sensing.  Not known to me works to use them for strain sensing.  In this sense the paper can be useful for the readers in the field.  However before the paper is accepted the following points should be addressed:

The strain simulation is unclear.  There is a discussion on the optics, the peak width and quality factor however this is well know.  The paper needs more discussion on the effect of the strain and what parameters of the grating it affects. How it is taken into consideration in COMSOL simulation and the EMT approach. It looks that the sensitivity does not depend on the direction of the strain and this is strange as strain perpendicular to the lines is expected to change the period more than along the lines. The authors missed some important papers on using the GMR as thermooptic sensor and sensor to refractive index in water and works demonstrating ultranarrow filters when the GMR is made of thin grating and waveguide such as: A. Sharon, D. Rosenblatt, A. A. Friesem, H. G. Weber, H. Engel, and R. Steingrueber, Opt. Lett. 21, 1564(1996).  I. Abdulhalim, Optimized guided mode resonant structure as thermooptic sensor and liquid crystal tunable filter, Chinese Optics Letters, 7 (8), 667, (2009). O. Krasnykov et.al, Optimizing the guided mode resonance structure for optical sensing in water, Physics Express 1(3), 183-190 (2011).  Latest work showed the existence of peaks with ultranarrow width that might be very useful for strain sensing: Sivan Isaacs et.al., Resonant Grating without a Planar Waveguide Layer as a Refractive Index Sensor, Sensors 19, 30003 (13p) (2019).

Reviewer 2 Report

Your paper entitled “Axially-Anisotropic Hierarchical Grating 2D 2 Guided-Mode Resonance Strain-Sensor” is very interesting.  The civil and structural engineering community could benefit from an optics-based strain sensor with an increased quality factor and sensitivity.  Your new slotted-disc Guided-mode resonance (GMR) grating design, which produces resonances with at least a 6-fold increase in quality factor over the original design with more axially-symmetric sensitivities, represents advancement over existing methods that require new materials to preserve resonance position. The slotting design permits the control of the duty cycle of the sub-grating, the lowering of the effective index of refraction of the high index part of the 2D GMR grating, and reducing the modulation index to increase the qualify factor.

Reviewer 3 Report

Babu and J.-B. Lee

This manuscript extends some previous work on patterned guided-mode resonant sensors by performing a series of numerical simulations in which the structures are modified such that the refractive index of the patterned disks is modulated. This variation reduces the effective index reduces the contrast between the patterned material and the substrate on which it is fabricated. This reduction leads to a small modulation index which is inversely proportional to the quality factor of the resonator. Consequently, the Q of the system can be increased thereby enhancing the sensitivity of the sensor platform.

I’m ambivalent on this manuscript because it appears to be a slight modification to previous work. However, if the following comments are considered the manuscript may be in a form suitable for publication.

In section 3.1 the authors refer to “Foland et al. [7]” This is the wrong reference; reference 7 is not Foland. Must correct.

In the same section the simulation is not 100% clear. Was the pitch varied to simulate the strain? Or did you include additional physics and let Comsol determine what the new pitch was due to the application of some external force? Must be clarified.

Often the authors use spectrums when speaking of multiple spectra. While technically correct it reads awkwardly, in part because spectra is more common. Suggest switching spectrums to spectra.

How does the choice of the Floquet periodic boundary condition influence the results? If the system were finite how would this impact the sensitivity of the GMR sensor? In real systems, would you expect uniform strain across the device? Are there edge effects to be concerned with in real structures? Should be clarified.

When discussing the Q for the different systems under varying conditions., are the results presented accurate? The dip at 15% strain is somewhat concerning. Is this real or a numerical artifact? Are the other fluctuations in the results of artifacts, or was this additional variation real? If it is real, what is the cause? Should be addressed.

Line 288: “By apply…” should be “By applying…” Must be corrected.

If these points are adequately addressed, the manuscript should be in a better position to be published. For now, the manuscript should be revised and resubmitted. At which point, I believe it would suitable for publication.

Round 2

Reviewer 1 Report

Following the authors response to the reviewers comments I think the paper can be published now. One minor comment, some of the references are missing information such as the journal name in ref. 22.